# Inactivation of PTEN and ZFHX3 in Mammary Epithelial Cells Alters Patterns of Collective Cell Migration

**DOI:** 10.3390/ijms24010313

**Published:** 2022-12-24

**Authors:** Ali Dayoub, Artem I. Fokin, Maria E. Lomakina, John James, Marina Plays, Tom Jacquin, Nikita M. Novikov, Rostislav S. Vorobyov, Anastasia A. Schegoleva, Karina D. Rysenkova, Julia Gaboriaud, Sergey V. Leonov, Evgeny V. Denisov, Alexis M. Gautreau, Antonina Y. Alexandrova

**Affiliations:** 1N.N. Blokhin Cancer Research Center of the Ministry of Health of the Russian Federation, 115478 Moscow, Russia; 2Ecole Polytechnique, Institut Polytechnique de Paris, 91120 Palaiseau, France; 3Moscow Institute of Physics and Technology, 141700 Dolgoprudny, Russia; 4Cancer Research Institute, Tomsk National Research Medical Center, Russian Academy of Sciences, 634009 Tomsk, Russia

**Keywords:** cell migration, epithelial-to-mesenchymal transition, partial EMT, vimentin, E-cadherin, adherens junctions

## Abstract

Whole exome sequencing of invasive mammary carcinomas revealed the association of mutations in *PTEN* and *ZFHX3* tumor suppressor genes (TSGs). We generated single and combined *PTEN* and *ZFHX3* knock-outs (KOs) in the immortalized mammary epithelial cell line MCF10A to study the role of these genes and their potential synergy in migration regulation. Inactivation of *PTEN*, but not *ZFHX3*, induced the formation of large colonies in soft agar. *ZFHX3* inactivation in *PTEN* KO, however, increased colony numbers and normalized their size. Cell migration was affected in different ways upon *PTEN* and *ZFHX3* KO. Inactivation of *PTEN* enhanced coordinated cell motility and thus, the collective migration of epithelial islets and wound healing. In contrast, *ZFHX3* knockout resulted in the acquisition of uncoordinated cell movement associated with the appearance of immature adhesive junctions (AJs) and the increased expression of the mesenchymal marker vimentin. Inactivation of the two TSGs thus induces different stages of partial epithelial-to-mesenchymal transitions (EMT). Upon double KO (DKO), cells displayed still another motile state, characterized by a decreased coordination in collective migration and high levels of vimentin but a restoration of mature linear AJs. This study illustrates the plasticity of migration modes of mammary cells transformed by a combination of cancer-associated genes.

## 1. Introduction

The understanding of molecular mechanisms responsible for tumor progression have greatly benefited from the next-generation sequencing (NGS) of tumors. Nowadays, whole exome sequencing (WES) represents a fast and reliable way to identify the majority of mutations affecting cancer-associated genes. It also paves the way to precision medicine by potentially establishing the prognosis and supporting therapeutic decisions based on the mutations that the tumor harbors. The most deadly phase of cancer is when tumor cells disseminate throughout the organism and seed distant metastases. Despite initial expectations to identify genes specifically promoting migration and invasion of tumor cells, it appeared that most cancer-associated genes promoting cell migration also promote proliferation in a pleiotropic and perhaps even coupled manner [1,2].

Our understanding of tumor cell dissemination has greatly improved and simultaneously complexified. EMT, where non-mobile polarized epithelial cells with developed cell–cell adhesions turn into highly mobile individual mesenchymal cells, has been observed in the progression of a number of carcinomas [3]. It is now clear, however, that this process is not limited to the manifestation of the two extreme states of cells, epithelial versus mesenchymal, but is rather multistep with a series of changes during which cells gradually lose their epithelial characteristics and acquire properties of mesenchymal cells. The main feature of full EMT is the complete downregulation of E-cadherin-mediated AJs. The early stages of EMT are not associated with a significant downregulation of E-cadherin but are accompanied by significant rearrangement of AJs [4,5,6,7]. Developed E-cadherin AJs are compact and appear as a single E-cadherin line, referred to as linear or tangential AJs, whereas disparate non-linear AJs can be represented by individual E-cadherin dots combined into a dotted line (punctuated AJs) or by radial stripes (radial AJs) [8]. Reorganization of the AJ indicates a weakening of cell–cell interactions and the acquisition of a mobile phenotype by cells. Cells retain cell–cell adhesions, the main characteristic of epithelial cells, while acquiring migratory abilities, in a process called partial or hybrid EMT [9,10]. Such partial EMTs lead to collective migration [11,12,13] and proved to be more capable of forming tumors than complete EMTs [14]. Carcinoma cells in the partial EMT state more efficiently metastasize than cells with a fully epithelial or mesenchymal phenotype [4,5,9,15]. Numerous cancer driver genes have been reported to enhance migration and invasion of tumor cells [16], yet how these genes cooperate to produce a given mode of migration has been poorly studied.

PTEN (Phosphatase and tensin homolog) is frequently deleted in many cancers, such as glioblastoma, breast, lung, colon and prostate cancer [17,18,19]. Hemizygous *PTEN* deletions were associated with poor prognosis of mammary carcinomas [17]. In addition to the classical loss of heterozygosity (LOH) in tumors, *PTEN* gene is also frequently silenced by epigenetic factors during tumor progression [17,20,21,22]. *PTEN* loss induced EMT in tumor cells of colon cancer [23], prostate cancer [24] and breast cancer [25,26]. PTEN inactivation was also reported to increase collective migration of non-tumor cells both in vitro and in vivo [27]. Downregulation of *PTEN* enhanced cell migration and wound healing of human corneal epithelial cell monolayers and rat cornea scratch wound models, as well as in the model of the wound healing of whole rat eyes [28].

ZFHX3 (Zinc finger homeobox protein 3), also known as ATBF1 (AT motif binding factor 1), is a transcription factor that was implicated in myogenic differentiation, cell proliferation and embryogenesis [29,30]. It is a TSG in multiple types of cancers, and its genomic deletion or mRNA downregulation promotes the prostate, breast, head and neck, and gastric cancer progression [31]. In epidermal keratinocytes, *ZFHX3* downregulation promotes cell migration and EMT traits, such as the loss of E-cadherin-mediated junctions and their replacement by N-cadherin-mediated junctions [32]. The transcription factor *ZFHX3* is rarely mutated, but its mRNA expression level is almost systematically reduced in breast cancer cell lines [33,34]. Moreover, low levels of ZFHX3 are strongly correlated with poor prognosis of breast cancer patients [34].

Herein, we studied the combination of these two TSGs (*PTEN* and *ZFHX3*) that we have found mutated simultaneously in the same invasive mammary carcinoma. To understand their contribution to cell transformation and migration, we inactivated these two genes in an immortalized, but untransformed, mammary epithelial cell line, MCF10A.

## 2. Results

### 2.1. Single and Combined KOs of Genes Found Altered in Mammary Carcinomas

We performed whole exome sequencing of 10 invasive breast carcinomas of no special type (classified according the 2019 World Health Organization classification of tumors of the breast) [35]. To determine driver mutations, we compared tumor and peripheral blood samples of the same patients and identified tumor-specific mutations. Then we identified putative driver mutations by comparing them with the list of cancer driver genes curated by the IntOGen database (Appendix A). Although PTEN suppression is shown in many tumors as we wrote above, the loss of *PTEN* as a driver mutation is found in 5.9% of tumors, according to the AACR Project GENIE database [36]. In our small tumor collection, it occurred as a driver mutation one time. In this case, it was combined with the ZFHX3 mutation (marked in purple in Appendix A). According to the AACR Project GENIE database [36], *ZFHX3* is mutated in 3.0% of tumors. Based on the TCGA database, we found that the combination of *PTEN* and *ZFHX3* mutations is present in 0.91% of cases (10 of 1098 cases). We also calculated the association of overall survival with both *PTEN* and *ZFHX3* mutation status using the cBioPortal tool. A correlation between *PTEN* genetic alterations and overall survival of breast cancer patients was detected. In contrast, *ZFHX3* mutations were not associated with survival prognosis (Appendix A). We pondered if the combination of these two driver mutations was indicative of a synergy between the two driver genes. Therefore we decided to study their potential synergy by deriving single and combined KOs in the MCF10A cell line, originating from a human fibrotic breast that is immortalized but not transformed [37]. *PTEN* KO in MCF10A was commercially available. We generated biallelic KO of *ZFHX3* in parental MCF10A and in the *PTEN* KO (Appendix A). Thus, we obtained a series of 4 cell lines: parental MCF10A, single KOs of *PTEN* and *ZFHX3* and the double KO of both genes (DKO). The evidence for successful *PTEN* and *ZFHX3* knockouts in the resulting *PTEN* KO, *ZFHX3* KO and DKO cell lines was verified by Western blot analysis (Appendix A).

### 2.2. Morphology Alterations Caused by PTEN and ZFHX3 KOs

MCF10A cells in tissue culture dishes display variable morphologies. Epithelial islets usually co-exist with isolated cells. PTEN KO cells exhibited pronounced epithelial phenotype with small islets and rare isolated cells (Appendix A). On the contrary, in ZFHX3 KO cultures, the epithelial morphology is less defined and more isolated cells are present. This effect of *ZFHX3* KO inactivation is reversed when combined with *PTEN* KO in DKO cells. The addition of horse serum and EGF (epidermal growth factor) stimulated scattering of islets and acquisition of mesenchymal phenotype (Appendix A), and vice versa the incubation of cells in depleted medium (without EGF and with only 1% horse serum) potentiated the epithelial phenotype in parental MCF10A cells. Our task was to analyze the contribution of *PTEN* and *ZFHX3* KOs to the acquisition of EMT and migration pattern of mammary epithelial cells. Therefore, we performed our experiments in depleted medium (Figure 1). In these conditions, parental MCF10A cells in sparse culture (no more that 50% of surface occupied by cells) formed islets, and isolated cells were practically absent. *PTEN* KO cells in depleted medium also formed islets with pronounced cell–cell adhesion between cells. *ZFHX3* KO formed loose islets consisting of well-spread-out cells, there were numerous small gaps between cells (red arrows in Figure 1A,C) and also some individual cells scattered from groups. DKO cells were aggregated in islets (Figure 1A). To evaluate cell–cell contacts in these cultures, we stained cells for E-cadherin and quantified the different types of E-cadherin Adherens Junctions (AJs) [8]. It is sometimes difficult to unambiguously attribute AJ morphology to a category, so we distinguished linear (tangential) AJs, which are typically epithelial, from non-linear AJs, which combine punctuated and radial contacts, and portrayed more plastic AJs. Parental MCF10A cells in a depleted medium formed typical linear AJs, tangential to cell boundaries of the tight islets (Figure 1A,B).

The main tendency was that both *PTEN* and *ZFHX3* KO induced significant rearrangement of AJs: the *PTEN* KO cells displayed mainly punctuate E-cadherin AJs looking like dashed lines, while *ZFHX3* KO caused the transformation of linear AJs into radial ones. DKO cells exhibited well-pronounced linear AJs similar to those in parental cells (Figure 1A,B).

We also estimated the proliferation activity of parental and gene-edited cell lines by examining the incorporation of EdU into nuclei of cells both in depleted medium and with addition of horse serum and EGF. The percentage of cells in S-phase was determined as a ratio of EdU-positive nuclei/DAPI-stained nuclei. We did not reveal significant differences in proliferation of parental and KO cells for 16 h of incubation in depleted medium. The horse serum and EGF stimulated significantly cell proliferation, but there were also no detected significant differences between lines (Figure 1D).

Since *PTEN* and *ZFHX3* were described as TSGs for many cancers, we wanted to evaluate the level of transformation that these genetic alterations provide to untransformed MCF10A cells. To this end, we tested the ability of the genome edited cells to grow colonies in soft agar. After 3 weeks of anchorage independent growth, parental MCF10A cells formed only rare and small colonies. In contrast, *PTEN* KO cells formed colonies of various sizes, some of them being very large (Figure 2).

*ZFHX3* KO cells formed more colonies than parental cells, but less than *PTEN* KO cells. All colonies of *ZFHX3* KO cells were of small size. When the two KOs were combined, colony size was intermediate, in between the sizes displayed by the two single KOs. However, the number of colonies increased, indicating some synergy in between the two TSGs on this parameter.

### 2.3. Migration of Epithelial Islets

To study the behavior of epithelial islets, we recorded the motility of cells in sparse cell cultures (when only about 50% of substrate is covered with cells) in the depleted medium. Cells within small islets consisting of 5–10 cells were tracked (Appendix A). Trajectories were plotted in a chart where the various cells are registered from the same origin at the center of the graph (Figure 3A). Trajectories highlighted that single *PTEN* KO cells in islets migrated much more actively than cells of another genotype. This is illustrated by the mean square displacement (MSD) that corresponds to the area explored per interval of time (Figure 3B). This higher exploration of *PTEN* KO cells was associated with increases in all parameters that contributed to migration efficiency, namely persistence, directionality and speed (Figure 3C–E). Even though *ZFHX3* KO cells were forming lamellipodia and were actively migrating, they were not coordinated and islets did not move as a whole, unlike *PTEN* KO islets. In DKO cells, *ZFHX3* inactivation suppressed to a large extent, but not completely, the increase in persistence, directionality and cell speed provided by *PTEN* inactivation.

### 2.4. Influence of PTEN and ZFHX3 KOs on Cell Motility inside a Monolayer

As a second approach for migration analysis, we used the experimental wound assay (Figure 4A). *PTEN* KO cells show enhanced healing ability compared to parental cells, while *ZFHX3* KO strongly inhibits the healing process (Figure 4A,B, Appendix A). The DKO displayed moderately enhanced wound healing compared to parental MCF10A. The velocity–magnitude heat map showed that *PTEN* KO strongly magnifies the speed difference between cells at the wound periphery and cells inside the monolayer, whereas *ZFHX3* KO equalizes front and rear speeds (Figure 4A,C). DKO cells displayed a front and rear speed difference similar to the parental cells.

To investigate migration coordination, we performed particle image velocimetry (PIV) to display cell flow inside the monolayer and at the wound periphery (Figure 4D). Flow fields for *PTEN* KO showed that cells migrate directionally to close the wound, while *ZFHX3* KO cells flow in multiple directions. DKO cells reveal an overall coordinated flow interrupted by some uncoordinated areas. To further understand cell coordination, heat maps of the order parameter, i.e., cosine of the angle of displacement vectors with the overall direction of wound closure, were built for each cell line (Figure 4E). In comparison with parental cells, *PTEN* KO enhanced the coordination of border cells, making them move more congruently with the overall direction of the wound closure. On the contrary, *ZFHX3* KO cells displayed an uncoordinated pattern of migration of border cells. An intermediate level of coordination was displayed by DKO cells.

Overall, these results showed that both *ZFHX3* and *PTEN* genes are crucial in modulating collective migration, but their inactivation results in opposite patterns of collective migration regulation. *ZFHX3* KO cells acquire an independent motility pattern with less coordination, whereas *PTEN* KO cells are highly coordinated, thus promoting wound healing.

### 2.5. Differences in Structure of Acini Formed by Parental and Gene-Edited Breast MCF10A Cells

One of the criteria which permit the estimate of the grade of malignancy of tumor cells is the morphology of the acini which these cells formed in 3D Matrigel [38]. We provide such analysis and discovered that *PTEN* KO or DKO greatly increased the acini size, whereas *ZFHX3* KO led to a decrease in acini diameter. *PTEN* KO exhibits a more elongated form of acini, which is a sign of the increased invasion potential of these cells (Appendix A).

### 2.6. Aquisition of EMT Caused by Knockouts of PTEN and ZFHX3 in Breast Cells

Next, we checked how the alterations of motility behavior caused by KOs of *PTEN* and *ZFHX3* are associated with other EMT characteristics. First, we analyzed by Western blot the expression of the EMT markers E-cadherin; N-cadherin; cytokeratins 5, 6, 8; and vimentin (Figure 5A,B). As an additional marker, which is more indicative of partial EMT, we used P-cadherin [39]. The KOs of both *PTEN* and *ZFHX3* and their combination did not affect E-cadherin levels. N-cadherin expression was significantly reduced in *PTEN* KO and remained practically unchanged in other cell lines (Figure 5A,B). Basal cytokeratins 5/6 were increased in *PTEN* KO and DKO cells. The expression of cytokeratin 8, which was typical of luminal epithelial cells, was also increased in *PTEN* KO but not in DKO cells. A small amount of vimentin, which is one of the main markers of EMT, was expressed by parental MCF10A cells in depleted medium (Figure 5A,B). *PTEN* KO decreased vimentin expression, whereas *ZFHX3* KO and DKO cells increased vimentin levels significantly (Figure 5A,B). P-cadherin levels did not change as a result of *PTEN* and *ZFHX3* KO, but slightly increased in DKO cells, in a non-significant manner (Figure 5A,B).

We paid special attention to the position of vimentin-positive (Vim+) cells, either inside or at the border of islets. In parental MCF10A, Vim+ cells often localized at the border of islets. The very rare Vim+ cells in *PTEN* KO cultures were also localized at the periphery of islets. In *ZFHX3* KO cultures, the number of Vim+ cells significantly increased both inside and at the periphery of islets. DKO exhibited an additional increase of Vim+ cells, most of which localized at islets’ periphery (Figure 5C,D).

We also checked the distribution of Vim+ cells in monolayers during wound healing. In parental and *PTEN* KO cultures, most of the few Vim+ cells were localized at the wound edge and only rare Vim+ cells were inside the monolayer (Figure 5E). *ZFHX3* KO and DKO led to an increase in Vim+ cells both inside monolayer and at the wound edge. To address the question of whether the position of Vim+ cells influences wound healing, we analyzed the position of Vim+ cells in relation with cell motility and wound closure. The presence of Vim+ cells at the wound border was frequently associated with enhanced speed of the Vim^−^ cells surrounding Vim+ cells (Appendix A), raising the possibility that Vim+ cells play a leading role towards Vim^−^ follower cells.

Using confocal microscopy, we checked the morphology of E-cadherin AJs in the border cells of monolayers upon wound healing. E-cadherin AJs at the wound edge are not as tight as in cells further away from the wound, because border cells are actively migrating. Nevertheless, at the wound edge, parental MCF10A cells display mostly linear cell–cell AJs; *PTEN* KO cells have condensed dotted lines of E-cadherin (as seen in islets, Figure 1A), while *ZFHX3* KO cells exhibit mostly radial AJ (Figure 5E). DKO cells exhibit well-pronounced linear AJs with some elements of radial morphology.

Thus, our analysis revealed that *PTEN* and *ZFHX3* knockouts and their combination induced EMT but at different levels. According to markers expression, *PTEN* KO exhibited transition to a more epithelial phenotype (low N-cadherin and vimentin, increased expression of both cytokeratins (CK)5/6 and CK8). *ZFHX3* KO cells acquired the most mesenchymal phenotype—significantly increased vimentin and reduced CK8. DKO cells had a very high vimentin level and increased level of basal cytokeratins CK 5/6 and exhibited a tendency towards enhanced P-cadherin expression. At the same time, *PTEN* KO cells display significantly increased group motility in islets and in wound healing experiments. *ZFHX3* KO cells, together with the increase of vimentin levels, displayed a dramatic change in the morphology of E-cadherin AJs and an increase in the motility of individual cells, but cells kept E-cadherin AJs. Therefore, *ZFHX3* KO cells were at a more advanced stage of EMT than *PTEN* KO but did not acquire complete EMT, and their motility was not coordinated in islets or inside the monolayer during wound healing. The migration efficiency of these cells on 2D substrate was reduced compared to that of *PTEN* KO cells.

## 3. Discussion

In this work, we have mimicked the inactivation of TSGs that naturally occurs in mammary carcinomas by knocking out these genes in untransformed mammary epithelial cells. *PTEN* loss activates the phosphatidylinositol 3-kinase (PI3K) pathway, since *PTEN* encodes the major phosphatase that dephosphorylates the phospholipid PIP3 into PIP2. Activation of the PI3K catalytic subunit by mutations of the *PIK3CA* oncogene is the most frequent alteration of mammary carcinomas [40,41,42]. *ZFHX3* encodes a transcription factor that controls mammary cell proliferation in response to estrogen signaling [43]. The *ZFHX3* gene is sometimes mutated but more often downregulated in mammary carcinomas [33]. We found genetic alterations of *PTEN* and *ZFHX3* in tumors of breast cancer patients. Previously, the same combined inactivation of *ZFHX3* and *PTEN* was found to drive the progression of prostate cancer in a mouse model [44]. The synergy was due to the enhanced activation of both PI3K and ERK MAP kinase pathways that promoted proliferation. In our MCF10A cell system, *PTEN* KO was having a more dramatic effect on anchorage-independent growth than *ZFHX3* KO. In these soft agar experiments, we detected a synergy in the number of colonies but not on their size.

Previously it was shown that *PTEN* was implicated in the control of cell migration. Its inactivation increases collective migration of tumor cells from colorectal cancer and of primary glial cells [23,27]. The latter effect was independent from its negative role in PI3K signaling. PTEN is a dual phosphatase that not only dephosphorylates the PIP3 phospholipid but also the Abi1 protein [45]. Abi1 is a subunit of the WAVE regulatory complex (WRC) that activates polymerization of branched actin at the leading edge of migrating cells [46]. When Abi1 phosphorylation is opposed by PTEN, Abi1 is degraded by the calpain protease and the WRC destabilized, thus providing a possible mechanism by which PTEN loss can promote cell migration [25,45]. In line with previous reports, we found that MCF10A cells knocked out for *PTEN* migrated more persistently in a collective manner. To our knowledge, *ZFHX3* was not previously reported to control cell migration. Herein, we report that *ZFHX3* KO resulted in profoundly uncoordinated collective migration, presumably through its dramatic effect on AJs. The combination of the two KOs indicates that each of the driver gene is dominant over the other for some characteristics but not for all of them (Figure 6).

The migration phenotypes we report for *PTEN* and *ZFHX3* KOs can be interpreted in the framework of partial EMT. We did not observe complete dissociation of MCF10A epithelial cells in association with mesenchymal migration that together would represent a complete EMT. We did not observe the alteration of E-cadherin expression after *PTEN* or *ZFHX3* knockouts but noticed remarkable reorganization of E-cadherin AJs associated with alteration of cell motility. Decrease of N-cadherin expression even in comparison with parental MCF10A was shown only for *PTEN* KO, which, together with vimentin depletion, indicated that these cells obtained a more epithelial phenotype. However, at the same time, exactly *PTEN* KO cells were the most efficient in wound healing, gave the growth of the biggest colonies in soft agar and formed the most elongated acini in Matrigel, which suggests that cells with this mutation acquired a more invasive potential [38]. Furthermore, we observed the increase in expression of basal cytokeratins 5/6 in *PTEN* KO and *DKO* cells, which was estimated to be evidence of acquisition of a ‘basal-like’ molecular phenotype and is associated with poor prognosis [47]. The combination of these acquired traits may be indicative of partial or hybrid EMT, and for each of our cell lines, this combination is unique, which means that these lines acquired the EMT of different levels. It was shown earlier that MCF10A cells are highly plastic and showed mesenchymal characteristics in sparse cultures, while in a monolayer, they looked like epithelium [48]. Indeed, when these cells grow in complete medium (with EGF and horse serum), they look like individual mesenchymal cells and show mesenchymal features. In our experiments, we specifically cultured the cells in depleted medium to stimulate the epithelial phenotype, so we think that, in our conditions, we could trace changes caused not by external stimulation with growth factors but by investigating the results of genes knockout.

The main goal of our work was to analyze the alterations in cell migration caused by *PTEN* and *ZFHX3* KOs. Both knockouts and their combination led to reorganization of AJs, making them weaker. As a result, the cells became more motile, but in the case of *PTEN* KO, they still maintained good cell–cell contacts and thus could move as a coordinated group.

Vimentin was downregulated in *PTEN* KO cultures, which behaved overall as a tight but highly motile epithelium, but its expression was retained in fast migrating cells at the wound edge, and exactly at these regions, we observed the enhanced collective migration [49,50]. The Vim+ cells at the border of islet or wound could be leader cells, and while they keep pronouncing cell–cell contacts with followers, they determine the direction of group migration. In *ZFHX3* KO cells, we observed a strong increase in vimentin expression associated with radial E-cadherin AJs, two unambiguous signs of partial EMT [6,51]. Thus, these cells became motile but their cell–cell contacts are weak, and they could not maintain motility coordination. As a result, cells move to different direction inside islets or the monolayer and did not show effective group migration. It was striking to observe linear AJs associated with high vimentin levels in DKO cells, a genuine hybrid phenotype and what determines successful migration, thought not as effective as in *PTEN* KO.

Herein, we have described in detail the complex and subtle modifications in the patterns of migration when two TSGs, *PTEN* and *ZFHX3*, were inactivated. In recent work, *PTEN* deletion was combined with Ras activation, which is frequently observed in tumors [52]. In this work, it was shown that expression of activated KRAS induced non-directional uncoordinated cell movement that suppressed the enhanced collective migration of *PTEN* KO cells, as *ZFHX3* KO did in our study. It is important that the combination of *PTEN* and *KRAS* genetic alterations led to appearance of aggressive tumor cells [53] and the development of lung metastasis [54]. These two examples of *PTEN* inactivation combined with either inactivation of *ZFHX3* or activation of *KRAS* illustrate how migration characteristics are controlled by cancer driver genes that are primarily well established to control cell proliferation and to which extent migration patterns are plastic.

We showed that, in our migration analysis, the proliferation impact was not significant. However, when we said about the development of metastasis, the proliferation activity of mutated tumor cells is an essential characteristic. In our gene-edited cells in experiments with colonies grown in soft agar and acini formation in Matrigel, the size of colonies and mammospheres of *PTEN* KO and DKO cells were significantly bigger. It means that, in a more prolonged condition, the proliferation speed of *PTEN* KO and DKO could make a significant contribution. It is important to notice that migration is strongly coordinated with proliferation [2]. Thus, the acquisition of partial EMT, stimulated migration could favor proliferation and facilitate colony formation in the new destination. On the basis of our research, we could say that EMT is not a straightforward process of transition from the epithelial to the mesenchymal phenotype, which leads to an increase in the invasive ability of tumor cells. Although *ZFHX3* KO cells look more advanced in EMT, they did not exhibit greater migration efficiency or a more invasive phenotype in the 3D Matrigel. Conversely, *PTEN* KO and DKO cells, which are more epithelial, showed not only a better migration ability but also looked like more invasive cells by 3D Matrigel assay. In the process of EMT, cells can acquire various new properties, some of which may give them advantages in their ability to migrate, while others may give them some other advantages or none at all. For example, it is now clear that, for successful migration, partial EMT is even better than full EMT [15]. Our cells exhibit different EMT characteristics, and it is not clear which ones significantly improve migration. We believe that performing such an analysis on different tumor cells is very important to understand which features of EMT should be considered for the prognosis of the disease and the search for its treatment.

## 4. Materials and Methods

### 4.1. Sequencing of Invasive Mammary Carcinomas and Putative Driver Mutations

Ten patients with invasive breast carcinoma of no special type were diagnosed and treated in the Cancer Research Institute of Tomsk NRMC (Tomsk, Russia). The histological diagnosis of breast cancer was made according to the WHO criteria [35]. All tumors were of luminal subtype, T_1–3_N_0–2_M_0_ stage, and the age range of patients was 41–67. DNA was isolated from fresh-frozen tumors (*n* = 10) and peripheral blood samples (*n* = 10) using a DNeasy Blood & Tissue kit (Qiagen sciences Inc., Germantown, MD, USA). DNA and library quality was measured using a 2200 TapeStation Instrument (Agilent Technologies, Santa Clara, CA, USA). Whole exome libraries were prepared using SureSelect XT Human All Exon v7 kit (Agilent, USA) and sequenced on a NextSeq 500 instrument (Illumina, San Diego, CA, USA) using paired-end 150 reads. Data were analyzed using the GATK pipeline, and genetic variants were annotated using the ANNOVAR [55,56]. Mutations that were present in peripheral blood were filtered out. Driver genes were identified based on the IntOGen database. Overall survival was calculated for breast cancer patients depending on PTEN and ZFHX3 mutation status using the cBioPortal tool.

### 4.2. Cell Culture

*PTEN* KO (homozygous deletion of exon 2, HD 101-006) and parental MCF10A cells were obtained from Horizon Discovery (Horizon Discovery Limited, Cambridge, UK). Parental MCF10A cells and their derivatives were cultured in DMEM/F12 medium (Gibco, Thermo Fisher Scientific, Waltham, MA, USA) supplemented with 5% horse serum (Hyclone, Cytiva, South Logan, UT, USA), 100 ng/mL cholera toxin (Sigma-Aldrich, St. Louis, MO, USA), 20 ng/mL epidermal growth factor (Sigma-Aldrich, USA), 0.01 mg/mL insulin (Sigma-Aldrich, USA), 500 ng/mL hydrocortisone (Sigma-Aldrich, USA) and 100 U/mL penicillin/streptomycin (PanEco, Moscow, Russia) (full media).

For Western blot, immunofluorescence and live-cell-imaging experiments, cells were seeded into dishes or chambers and cultured in full media overnight, then the medium was changed to DMEM/F12 medium (Gibco, USA) supplemented with 1% horse serum (Hyclone, USA), 100 ng/mL cholera toxin (Sigma-Aldrich, USA), 0.01 mg/mL insulin (Sigma-Aldrich, USA), 500 ng/mL hydrocortisone (Sigma-Aldrich, USA) and 100 U/mL penicillin/streptomycin (depleted media without EGF and with 1% horse serum).

After 24 h of incubation in depleted medium, live cell imaging was performed. For live imaging, we used depleted DMEM/F12 without phenol red (Sigma-Aldrich, USA), supplemented with 15 mM HEPES and 1% horse serum. After imaging, cells were fixed and stained for immunofluorescence or lysed for Western blot experiments.

### 4.3. Generation of KO Cell Lines

*ZFHX3* KO were obtained in parental or *PTEN* KO lines using CRISPR/Cas9 and the following targeting sequence: 5′-TCGTCTCGGGGAAGGACAAT-3′ (CRISPR30279_CR, Thermo Fischer Scientific, Waltham, MA, USA), that covers all known isoforms of ZFHX3. Cells were transfected with crRNA:tracrRNA duplex and the purified Cas9 protein by Lipofectamine CRISPRMAX™ Cas9 transfection reagent (all reagents were from ThermoFisher). Cells were then diluted to 0.8 cells/well into 96-well plates. The presence of mutations was evaluated by direct Sanger sequencing of genomic PCR product. Clones displaying biallelic frameshift mutations were selected for the experiments.

### 4.4. Cell Proliferation Analysis

Proliferation analysis was performed as described in Molinié et al., 2019 [2]. Briefly, the cells were seeded on 24-well plates at 2.5 × 10^4^ cells per well. Cells were deprived of serum for 36 h and stimulated with the indicated media for 16 h. Moreover, 1% GF media corresponds to 1% horse serum and 4 ng/mL of EGF. EdU was added to the cells 1 h before fixation and processed with a Click-it EdU Imaging kit (Thermo Fisher Scientific, #C10337). The percentage of cells in S-phase was determined as a ratio of EdU—positive nuclei/DAPI-stained nuclei. The data represent three independent experiments with SEM. Measurements were performed from six different fields of view (N = 600−1000). Statistical analysis was performed by Kruskal–Wallis test

### 4.5. Anchorage-Independent Cell Growth in Soft Agar

For the estimation of capacity of gene-edited cells for anchorage-independent growth, we used protocol suggested by Du et al. [57]. A total of 5000 cells per well of a 12-well plate were embedded in 0.5 mL of 0.35% agar, prepared on full MCF10-A medium. A total of 0.5 mL of 0.8% agar, made on full MCF10A medium, was used as a bottom layer. During 3 weeks of cell growth, 0.5 mL of fresh medium was added on the top of soft agar every 4–5 days. Pictures of cell colonies were taken on the Olympus CKX53 microscope supplied with the DP22 camera (Olympus, Tokyo, Japan). Colony size was analyzed using the Fiji software (freehand selection tool). Results are expressed as the means and standard errors of the mean (SEM) from three independent experiments. Statistical analysis was performed in the GraphPad software using ANOVA test.

### 4.6. Acini Assay Method

MCF10A cells were seeded on top of polymerized Matrigel (CB-40230C, Corning) in a Millicell EZ SLIDE 8-well glass chamber slide (PEZGS0816, Millipore) in reduced EGF (4 ng/mL) and serum (1%) medium supplemented with 2% of Matrigel. For 3 weeks, medium was regularly changed. Acini were then fixed in 2% PFA in PBS permeabilized with 0.5% Triton X-100, rinsed with PBS/glycine (130 mM NaCl, 7 mM Na_2_HPO_4_, 3.5 mM NaH_2_PO_4_, 100 mM glycine) and blocked in IF Buffer (130 mM NaCl, 7 mM Na_2_HPO_4_, 3.5 mM NaH_2_PO_4_, 0.1% bovine serum albumin, 0.2% Triton X-100, 0.05% Tween-20) + 10% FBS first and then with IF Buffer + 10% FBS + 20 µg/mL goat anti-Rabblit Fc fragment (111-005-046, Jackson ImmunoResearch). Acini were incubated with the primary antibodies in the secondary block solution washed with IF buffer and then incubated with secondary antibody in IF Buffer + 10% FBS. Then acini were incubated with DAPI, rinsed with IF buffer and mounted with Abberior Mount Liquid Antifade (Abberior) and sealed with nail polish.

Images of acini were obtained on the SP8ST-WS confocal microscope equipped with a HC PL APO 40×/1.10 W CS2 water immersion objective, a white light laser, HyD and PMT detectors. Image analysis was performed in FIJI. Acini were manually outlined at their maximal projection; their size and aspect ratio were then analyzed. Data are mean ± s.e.m of two technical repeats with *n* = 22–23. Results were statistically analyzed by Kruskal–Wallis test.

### 4.7. Antibodies

For immunofluorescence microscopy, mouse monoclonal anti-E-cadherin, clone 36 (BD Transduction Laboratories, Franklin Lakes, NJ, USA), monoclonal anti-vimentin V9 antibodies (Sigma-Aldrich, USA) and anti-Laminin V, clone D4B5 (Merck) were used as primary antibodies. Goat polyclonal anti-mouse IgG1 conjugated with AlexaFluor488, goat polyclonal anti-mouse IgG2a conjugated with AlexaFluor 647 (Molecular Probes Inc., Eugene, OR, USA) and goat anti-mouse antibody conjugated with Alexa Fluor 647 from (Life Technologies) were used as secondary antibodies. Phalloidin 488 from (TebuBio) was added to the secondary antibodies. Nuclei were stained with DAPI (Sigma-Aldrich, USA).

For Western blot analysis and immunostaining, the following antibodies were used: mouse monoclonal anti-E-cadherin clone 36, mouse monoclonal anti-N-cadherin clone 32, mouse monoclonal anti-P-cadherin (BD Transduction Laboratories, USA); mouse monoclonal anti-vimentin V9 antibodies; mouse monoclonal anti-β-actin, clone C4; rabbit polyclonal anti-ZFHX3, clone ab119909, (Abcam); rabbit polyclonal anti-PTEN, clone 9552 (Cell Signaling); mouse monoclonal anti-GAPDH, clone, AM4300 (Invitrogen, Waltham, MA, USA); mouse monoclonal Anti-pan-Cytokeratin (Sigma-Aldrich, USA). Horseradish peroxidase-conjugated goat polyclonal anti-mouse and anti-rabbit IgG antibodies (Jackson ImmunoResearch, Ely, UK) were used as secondary antibodies. Other reagents were obtained from Sigma-Aldrich, USA. All antibodies and dyes were used in the concentrations recommended in their product specification sheets.

### 4.8. Immunofluorescence

For triple staining for E-cadherin, vimentin and DNA, cells were fixed with 1% paraformaldehyde prepared on DMEM/F12 with 20 mM HEPES at room temperature for 15 min and permeabilized with ice-cold methanol for 3 min at −20 °C. The mounted samples were examined with a Leica TCS SP5 confocal laser scanning microscope using the HDX PL APO 100× objective or with a Nikon Eclipse Ti-E microscope using the Plan Fluor 20× objective and ORCA-ER camera (Hamamatsu Photonics, Hamamatsu, Japan) controlled via NIS-Elements AR 3.22 software (Nikon, Tokyo, Japan) or with an Axioplan Zeiss epifluorescence microscope using a Plan-Neofluar 100×/1.3 lens (Carl Zeiss, Aalen, Germany). Vimentin-positive (Vim+) cells were counted in ImageJ software using the ‘cell counter’ plugin; E-cadherin AJs and intercellular gaps were scored manually. Data were from three independent experiments. Statistical analyses were performed using GraphPad software with a contingency Chi square test for E-cadherin junctions and Vim+ cells counting and an unpaired *t*-test for normalized gap counting.

### 4.9. Western Blot

Cells were washed twice with wash buffer (10 mM Tris-HCl pH7.5, 0.5 mM EDTA, 150 mM NaCl) and lysed with lysis buffer (wash buffer supplemented with 0.5% Na deoxycholate, 1% NP-40, protease inhibitor cocktail (La Roche Ltd., Basel, Switzerland), and phosphatase inhibitor cocktail (Sigma-Aldrich, USA)). Samples were mixed with 5× sample buffer (250 mM Tris-HCl pH 6.8, 10% SDS, 30% glycerol, 5% β-mercaptoethanol, 0.02% bromophenol blue), heated for 10 min at 95 °C and loaded onto SDS-polyacrylamide gel in equal protein concentrations according to the SDS-PAGE Bio-Rad protocol. Resolved proteins were transferred to Amersham Hybond-Cnitrocellulose hybridization membranes, 0.45 micron (GE Healthcare, Chicago, IL, USA). Membranes were blocked with 5% *m/v* bovine serum albumin solution (PanEco, Russia) for further staining of vimentin or with 5% nonfat milk (AppliChem, Darmstadt, Germany) for further staining of other proteins, in Tris-buffered saline with 0.1% *v/v* of Tween 20 (AppliChem, Germany) for 1 h followed by incubation with the primary antibodies at 4 °C overnight. After washing, peroxidase-conjugated secondary antibodies were applied for 1 h at room temperature. Blotted protein bands were detected using Pierce ECL Western Blotting Substrate (ThermoFisher Scientific, Waltham, MA, USA), and chemiluminescence images were captured by Image Quant LAS4000 (GE Healthcare, USA). The densitometry of results was performed in ImageJ software, and the results present means with standard errors of the mean (SEM) from three independent experiments. Statistical analysis was performed in GraphPad software using an unpaired *t*-test.

### 4.10. Cell Migration

All live imaging experiments were performed using a Nikon Eclipse Ti-E microscope equipped with the Nikon Plan 10× and Plan Apo 20× objectives and an ORCA-ER camera (Hamamatsu Photonics, Japan) controlled via NIS-Elements AR 3.22 software (Nikon, Japan).

#### 4.10.1. Migration of Islets

For the analysis of islets migration in sparse culture, cells were seeded in four-well cultivation chambers with glass bottom 155383 (Lab-Tek Thermo Fisher Scientific, New York, NY, USA). The time-lapse imaging of cells was performed 24 h after incubation with the depleted medium. Images were acquired every 10 min for ~16 h. Cell trajectories were obtained by tracking cells with ImageJ and analyzed using the DiPer software [58] to obtain the migration parameters: directional autocorrelation, mean square displacement, average cell speed and single-cell trajectories plotted at the origin. Data from three independent experiments were pooled for the analysis and plotted. The results are expressed as the means and standard errors of the mean (SEM). For migration persistence, a statistical analysis was performed using R. Persistence, measured as movement autocorrelation over time is fit for each cell by an exponential decay with a plateau, as described in [59].
$$A=\left(1-A_{min}\right)*e^{-\frac{t}{\tau }}+A_{min}$$
where *A* is the autocorrelation, *t* the time interval, *A_min_* the plateau and τ the time constant of decay. The plateau value *A_min_* is set to zero for cell lines in vitro as they do not display overall directional movement. The time constant *τ* of exponential fits was then compared using the Kruskal–Wallis test. The average cell speed and directionality were statistically analyzed using GraphPad software with the Kruskal–Wallis test.

#### 4.10.2. Wound Healing

To analyze collective cell migration, the wound healing assay was performed using the 2-well silicon inserts (80209, Ibidi GmbH, Grafelfing, Germany) into glass-bottomed dishes. Cells with a concentration of approximately 3× 10^5^ cells/mL were seeded inside each well of the insert with the full medium and cultivated overnight. Then, medium was changed to the depleted one (-EGF, 1% horse serum). After 24 h of incubation in the depleted medium, inserts were excluded and live-cell-imaging was performed. Images were acquired every 10 min for ~10 h until full wound closing. Wound closure percentages (mean with SEM) were calculated in parental and KOs cells from three independent experiments and statistically analyzed with the Kruskal–Wallis test using GraphPad software.

### 4.11. Analysis of Collective Cell Migration Using PIV

Monolayer heat maps of velocity magnitudes and flow direction inside the monolayer were built after performing PIV analysis using the open-source code PIVLab software on MatLab [60,61]. In brief, live-cell-imaging movies of wound edge (with time lapse 10 min) were processed on the PIVLab software using FFT window deformation algorithm with interrogation window of 128 × 128 px (108.8 micron × 108.8 micron), followed by two interrogation passes of 64 × 64 px and 32 × 32 px in the second and third passes, respectively, with an overlap of 50%. Erroneous vectors (shadows) were then filtered using image-based validation filters with a threshold (0.005). Average velocity values (means with SEM) of posterior and frontal cells were extracted over 7 h of migration from three independent experiments, and the results then were statistically analyzed in GraphPad software with a Mann–Whitney test between front and rear cells in parental and DKO conditions and an unpaired *t*-test between front and rear cells in PTEN KO and ZFHX KO conditions.

Following PIV analysis, order parameter local mapping was performed by the AVeMap software [62]. Order parameter was defined as a cosine of the angle between each local velocity vector and a normal drawn to the border of a wound. The cell monolayer was divided into 32 μm bars, starting from the edge of a wound. The meaning of all cosine values were averaged within each bar and were plotted in the form of a color kymograph.

## Figures and Tables

**Figure 1 ijms-24-00313-f001:**
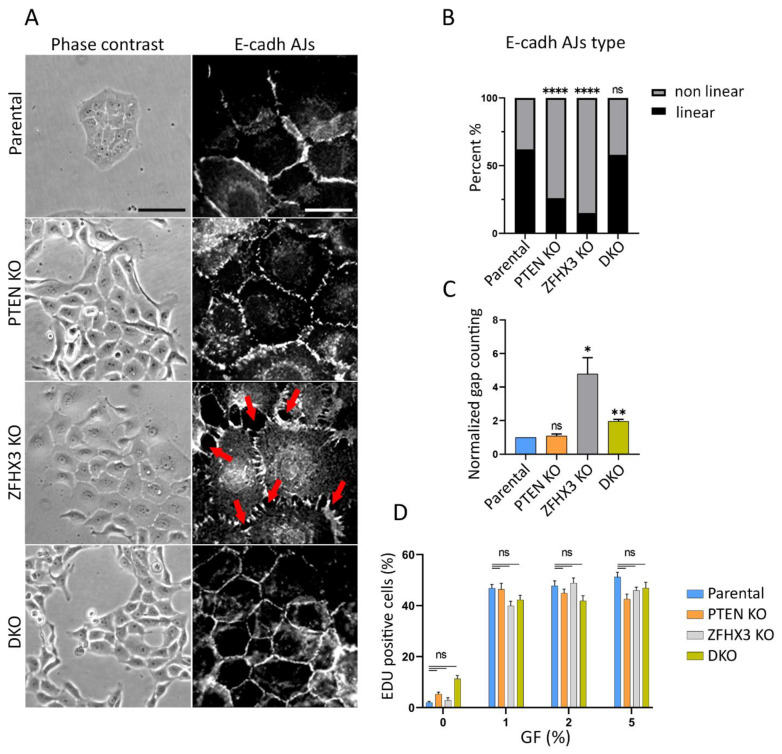
Effect of *PTEN* and *ZFHX3* knockouts on morphology, cell–cell adherences and proliferation activity of MCF10A cells. (**A**) Phase contrast view and E-cadherin AJs (immunofluorescence) of cells cultivated in depleted medium (1% of horse serum) and no EGF (epidermal growth factor). Red arrows point to gaps between cells. Scale bars 100 μm for phase contrast pictures and 20 μm for immunofluorescence. (**B**) Quantification of E-cadherin linear and non-linear AJs within islets of parental and KOs cells. (**C**) Quantification of gaps between cells in cell islets. The data (**B**,**C**) represent 3 independent experiments, Sample size (the number of calculated AJs) for each experiment is: Parental 155, 114, 199; *PTEN* KO 86, 87, 159; *ZFHX3* KO 98, 91, 161; DKO 136, 116, 189 junctions for (**B**,**C**). (**D**) Analysis of proliferation of obtained cells. The percentage of cells in S-phase was determined as a ratio of EdU—positive nuclei/DAPI-stained nuclei. The data represent 3 independent experiments. Measurements were performed from 6 different fields of view (*N* = 600−1000). Statistical analysis was performed by a contingency Chi square test in (**B)**; unpaired *t*-test in panel (**C)**, Kruskal–Wallis test in panel (**D**); * *p* < 0.05, ** *p* < 0.01, **** *p* < 0.0001 and ns: non-significant.

**Figure 2 ijms-24-00313-f002:**
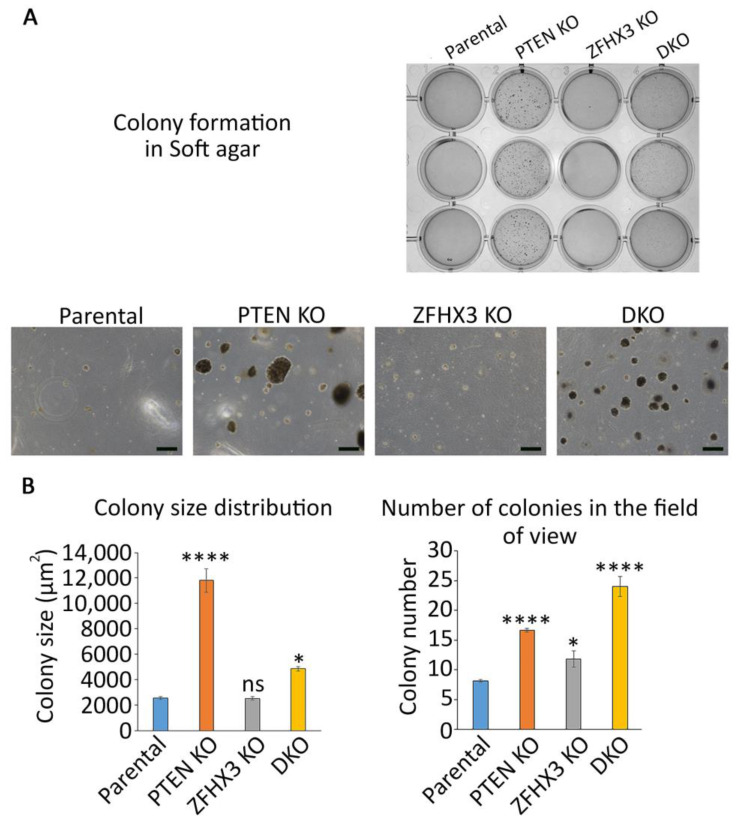
Anchorage independent growth of the gene-edited MCF10A cell lines in soft agar. (**A**) Overall view. Scale bars 200 μm. (**B**) Quantification of size (measured as apparent area) and numbers of colonies. *N* = 3, ANOVA, * *p* < 0.05, **** *p* < 0.0001 and ns: not significant.

**Figure 3 ijms-24-00313-f003:**
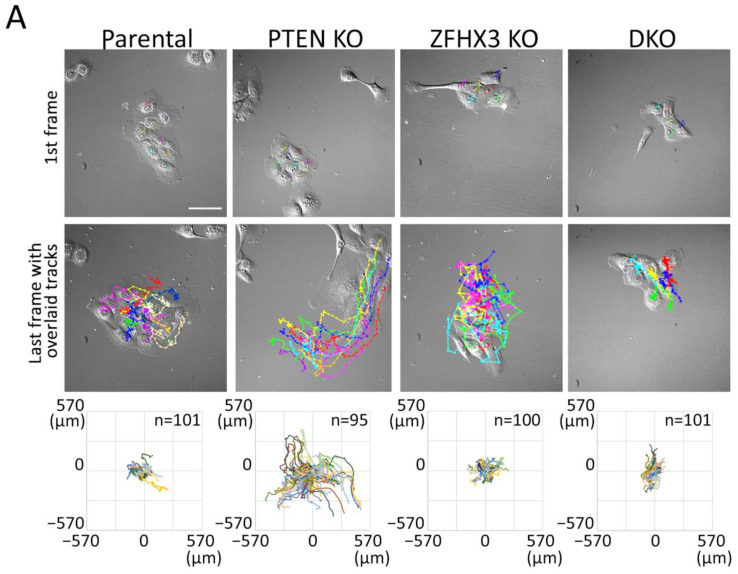
Collective migration of gene-edited MCF10A cell lines in sparse culture. (**A**) First frames of video and last frames with cell trajectories. Scale bar: 100 μm. (**B**–**E**) Plots characterized the migration behavior of gene-edited MCF10A cells: (**B**) A log-log plot of mean square displacements (MSD); (**C**) migration persistence; (**D**) directionality ratio over elapsed time (16 h); (**E**) mean cell speed. Data from 3 independent experiments; more than 100 cells for each line were monitored. Mean ± SEM. Kruskal–Wallis test, * *p* < 0.05, ** *p* < 0.01, *** *p* < 0.001, **** *p* < 0.0001 and ns: non-significant.

**Figure 4 ijms-24-00313-f004:**
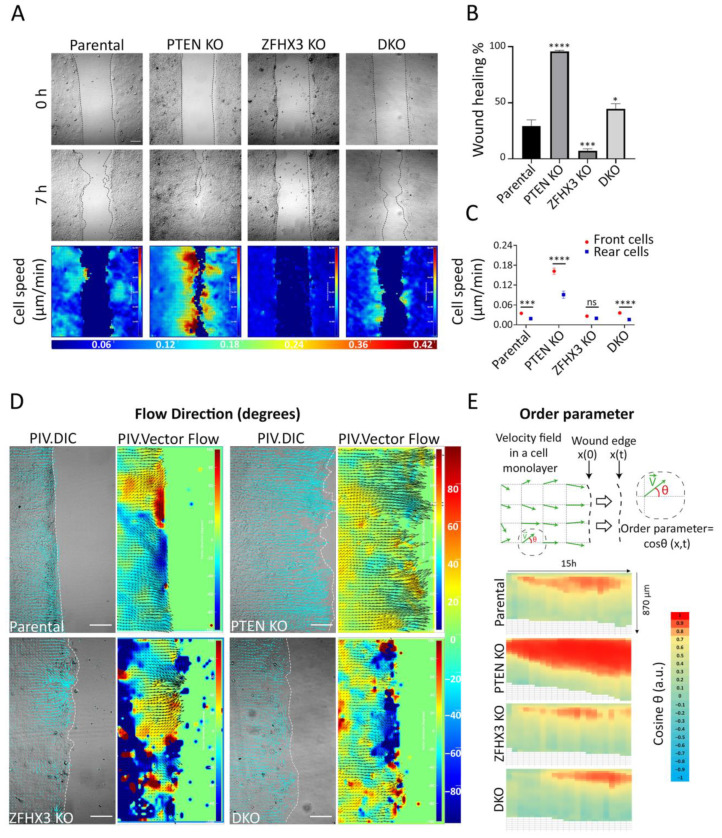
Wound healing of gene-edited MCF10A cell lines. (**A**) Heat map representation of cell speed after 7 h of migration. Scale bars: 100 μm. (**B**) Wound closure percentage after 7 h. (**C**) Cell speed; for front and rear cells in a monolayer. (**D**) Heat map representation of cell angles relative to the wound extracted from PIV analysis along the edge at the last hour of migration. Scale bars: 100 μm. (**E**) Local parameter order. Heat map of the average cosine value of the angle of displacement vectors over time. Data from 3 independent experiments. Total sample size is more than 15 videos for each cell line; mean ± SEM. Kruskal–Wallis test in Panel (**B**). Mann–Whitney test for parental and DKO cell lines, unpaired *t*-test for *PTEN* KO and *ZFHX3* KO cell lines in panel (**C**), * *p* < 0.05, *** *p* < 0.001, **** *p* < 0.0001 and ns non-significant.

**Figure 5 ijms-24-00313-f005:**
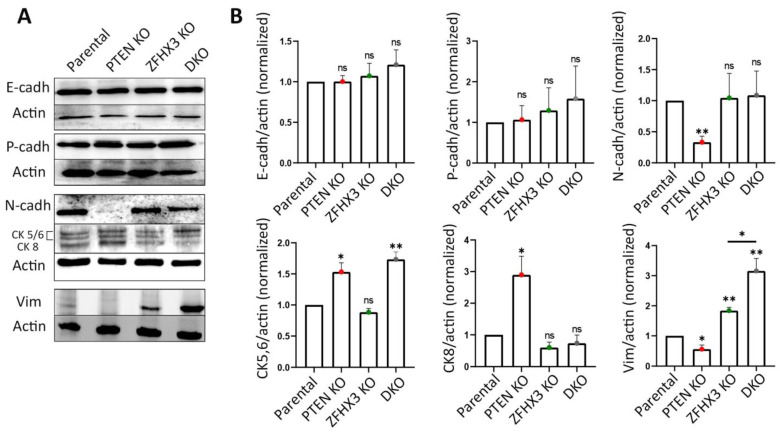
EMT features in gene-edited MCF10A cell lines. (**A**) Alteration of expression of E-, N- and P-cadherins; vimentin; and cytokeratins (5, 6, 8), and actin was used for normalization of protein loading (Western blots). (**B**) Quantification of Western blots by densitometry. Mean ± SEM from 3 independent experiments. (**C**) Immunofluorescence staining of islets for vimentin (green), E-cadherin (red) and nuclei (blue). Scale bars: 50 μm for islet images. (**D**) Quantification of vimentin-positive and -negative cells within islets or at their periphery. (**E**) Top images: immunofluorescence staining of wound edges for vimentin (green), E-cadherin (red) and nuclei (blue). Scale bars: 100 μm. Middle images: confocal microscopy of E-cadherin immunofluorescence in wound healing experiments. Nuclei are in cyan. Note the radial AJ in *ZFHX3* KO. Arrows indicate the magnified junctions displayed below. Scale bars 18 μm for main immunofluorescence images and 5 μm for magnified junctions. Three independent experiments. Sample size (the number of calculated Vim^+^ cells) for each experiment is: parental 193, 236, 438; *PTEN* KO 190, 235, 481; *ZFHX3* KO 150, 264, 424; DKO 250, 289, 356 cells for (**D**). Statistical analysis: unpaired *t*-test in panels (**B**); contingency Chi square test in (**D**); * *p* < 0.05, ** *p* < 0.01, **** *p* < 0.0001 and ns: non-significant.

**Figure 6 ijms-24-00313-f006:**
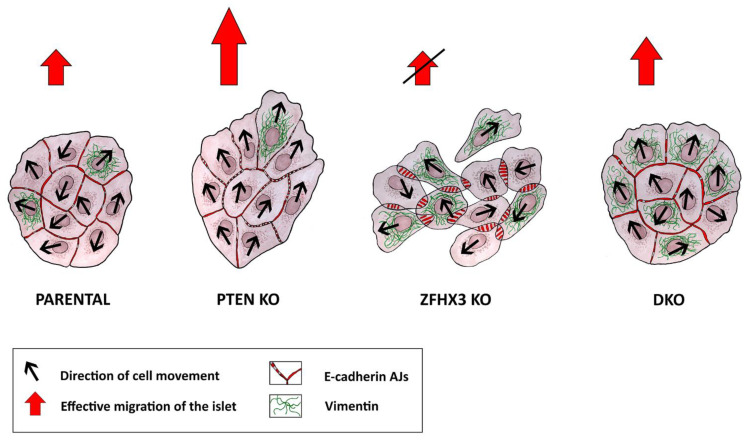
Plasticity of migration modes of genetically edited MCF10A cell lines. Knockouts of different TSGs induced partial EMT of different stages, which was associated with reorganization of AJs. MCF10A parental cells showed well-defined linear (tangential) AJs and demonstrated limited cell motility of islets. *PTEN* knockout was accompanied by the appearance of punctuated AJs followed with enhanced coordinated collective migration of epithelial islets. *ZFHX3* knockout resulted in the appearance of radial AJs and increased expression of the mesenchymal marker vimentin, which led to the weakening of cell–cell contacts and was accompanied by the loss of coordination of cell movement in islets and, as a result, a decrease in the efficiency of cell migration. DKO cells exhibited a different motile state, intermediate in terms of collective migration, characterized by a combination of linear AJs and high levels of vimentin.

## Data Availability

Not applicable.

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
