# Peer review of "Inactivation of PTEN and ZFHX3 in Mammary Epithelial Cells Alters Patterns of Collective Cell Migration"

_ijms, 2022, doi:10.3390/ijms24010313_

Round 1

Reviewer 1 Report

The authors study the effect of mutations in PTEN and ZFHX3 tumor suppressor genes, starting by establish the association of these genes by whole exome sequencing of invasive mammary carcinomas. They further explored the effect of these mutations in vitro by generation of single and combined PTEN and ZFHX3 knock-outs in MCF10A. Their results point to inactivation of these genes affect colony formation, cell migration patterns and modulation of adhesive junctions and vimentin expression. The authors conclude that the inactivation of the two genes induce different stages of partial EMT.

The study is overall well performed but some important issues need to be address.

Major issues

Line 37 – I would take “of cancers” out. And replace “which reveals prognosis and suggest treatments of cancers” with by potentially establishing the prognosis and supporting therapeutic decisions “based on the mutations that the tumor harbors”;

Line 54-55 – REF missing;

Line 86 – REF missing;

Line 92- “stoma” you mean stroma? Did you have a comparison with normal tissue or PBMCs? Or how did you reach this conclusion? This should be mentioned in the main text;

The bioinformatic results from 2.1. should be added as supplementary figures;

Line 98 – In the TCGA data, please analyze the frequency of both mutations being present, and if these patients have better or worse prognosis than single mutation patients;

Line 117 - The reasoning for adding or not EGF should be added in the main text;

In figure 1 did you try to quantification of the islet size or the size or number of gaps? To see if there is a significant difference? Might be worth adding these analyses. Otherwise you cannot state that the differences were not significant as in line 122.

Seems like the gaps between the DKO are also present but smaller/ less frequent?

Did you also check for the e-cadherin in full medium?

How do you explain that the DKO looks more similar to the parental cells? This should be explained in the discussion section;

On the assays from figure 2, why did you not add the crystal violet, as recommended in a typical clonogenic assay? Did you add some other dye? Please provide the reference for the protocol you used. Without a staining you might be missing some colonies;

Line 200 – I would not say controlling but rather modulating, displaying opposite effects. If PTEN is KO increased migration is observed and the opposite with ZFHX3 KO, which impairs migration;

Is a bit contradictory how ZFHX3 KO acquire more EMT characteristics and exhibits less migratory capacity, compared to PTEN is KO. How do you explain this?

The understanding of these phenotypes in metastasis formation, in in vivo models should be suggested as discussed as a future perspective in the discussion;

Line 331 – what do you mean by no “special type”? You mean histological subtype? Because molecular subtype, you describe between parenthesis, so is not clear to me what you mean;

Line 377 – please mention the Fiji function used for the colony counting and size analyses;

In 4.6. I did not quite understand how did you count the E-cadherin AJs with the cell counter plugin, could you please explain better?

Minor issues

Line 71 and 72 – “in vitro and in vivo “ should be in italic. Correct throughout the text;

Line 91 – Please simplify this phrase e.g.: “We performed whole exome sequencing of 10 invasive mammary carcinomas.”;

Line 96 – replace “wondered” with something more formal, such as, pondered;

Legend figure 1 – Add also EGF abbreviation, like the HS;

Line 126 – Change “control” to parental;

Figure 2 – Add a more descriptive text for example in B;

Line 148 – “consisting of”;

Legend fig 3, again describe a bit more and capitalize the sentences;

Line 166 – gene-edited instead of “genome-edited”;

Figure 4 – Capitalize Cell Speed (in A and C), velocity (in E) and correct parenral to parental (in B and C);

In figure 5B is there any specific reason for the significance not being represented as in all other graphs? Uniformize with *.

5D capitalize Vimentin;

Line 239 – “(…) lead to an increase (…)”;

Line 271 – “PTEN inactivation activates” please replace one of the words;

Line 288 – “(…) is a dual (…)”;

Line 300 – gene-edited no “genome-edited”;

In the methods, add the cities and countries of all the companies for reagent purchase;

Line 348 - penicillin/streptomycin purchase company info missing;

Line 363-366 – uniformize the font;

4.5. mention the concentrations of the antibodies;

In 4.6. the same antibodies are used? Please mention it, as well as the concentrations used;

In 4.7. the same antibodies are used? Please mention it, as well as the concentrations used;

Line 429 – add 4.8.1;

Line 450 – add 4.8.2.

Author Response

Answer to reviewers.

We are very grateful to the reviewers for their careful reading of our manuscript and their valuable comments. We believe that the quality of our paper has greatly improved after taking into account the reviewers' comments. Our responses and comments can be found below (marked as red).

Reviewer 1.

Major issues

Line 37 – I would take “of cancers” out. And replace “which reveals prognosis and suggest treatments of cancers” with by potentially establishing the prognosis and supporting therapeutic decisions “based on the mutations that the tumor harbors”; Done

Line 54-55 – REF missing; We added REF

Line 86 – REF missing; These data were going from our analysis of breast tumors, we re-phrased this sentence to make it more clear

Line 92- “stoma” you mean stroma? Did you have a comparison with normal tissue or PBMCs? Or how did you reach this conclusion? This should be mentioned in the main text;

We compared tumor and peripheral blood samples of the same patients.

To answer questions about bioinformatics we re-checked attentively our data and renew database analysis using more recent programs. The results of new analysis are in Table S1. Now it is written:

“Ten patients with invasive breast carcinoma of no special type were diagnosed and treated in the Cancer Research Institute of Tomsk NRMC (Tomsk, Russia). The histological diagnosis of breast cancer was made according to the WHO criteria Tan et al, 2020, https://doi.org/10.1111/his.14091 . All tumors were of luminal subtype, T1-3N0-2M0 stage, age range of patients was 41–67. DNA was isolated from fresh-frozen tumor (n=10) and peripheral blood samples (n=10) using DNeasy Blood & Tissue kit (Qiagen, USA). DNA and library quality was measured using 2200 TapeStation Instrument (Agilent, USA). Whole exome libraries were prepared using SureSelect XT Human All Exon v7 kit (Agilent, USA) and sequenced on a NextSeq 500 instrument (Illumina, USA) using paired-end 150 reads. Data were analyzed using the GATK pipeline, and genetic variants were annotated using the ANNOVAR. Mutations that were present in peripheral blood were filtered out. Driver genes were identified based on the IntOGen database.

The bioinformatic results from 2.1. should be added as supplementary figures; We added bioinformatics results to Supplementary data as Table S1

Line 98 – In the TCGA data, please analyze the frequency of both mutations being present, and if these patients have better or worse prognosis than single mutation patients;

We calculated the association of overall survival with both PTEN and ZFHX3 mutation status. The corresponding text and Figure S1 was added. Overall survival of breast cancer patients with double PTEN and ZFHX3 mutations was not calculated due to technical limitations of the cBioPortal tool.

Line 117 - The reasoning for adding or not EGF should be added in the main text;

It is main text.

It is written there: “The addition of horse serum and EGF stimulated EMT and vice versa the incubation of cells in depleted medium (without EGF and with only 1% horse serum) potentiated the epithelial phenotype in parental MCF10A cells. Our task was to analyze the contribution of each mutation to the acquisition of EMT and migration pattern of mammary epithelial cells. Therefor we performed our experiments in depleted medium.”

In figure 1 did you try to quantification of the islet size or the size or number of gaps? To see if there is a significant difference? Might be worth adding these analyses. Otherwise you cannot state that the differences were not significant as in line 122.  Seems like the gaps between the DKO are also present but smaller/ less frequent? We quantified the number of gaps inside islets for all our cells and added these results as Figure 1C

Did you also check for the e-cadherin in full medium? No, we performed all experiments in depleted medium. We moved the picture of cell morphology in full medium (5% horse serum+EGF) to Supplementary data (Figure S3), because further we did all experiments in depleted medium, and add the plot with analysis of proliferation of these cells in different conditions, because the rate of proliferation is important characteristic of obtained cells with mutations.

How do you explain that the DKO looks more similar to the parental cells? This should be explained in the discussion section;

On the base of our results we could speculate that while each single mutation (PTEN and ZFHX3 KOs) lead to acquisition of different and partially even opposite features of EMT, their combination lead not to simple amplification of EMT characteristics, but to their complementarity and acquisition of new EMT signs (as significant increase of vimentin expression). In results we could see that in some characteristics DKO cell resembled parental MCF10A (linear AJs, formation of dense islets). At the same time we could see significant increase of expression of cytokeratins 5,6, what is estimated as sign of poor prognosis for breast tumors, increased wound healing, formation of numerous colonies in soft agar and significant increase of size of acini in matrigel. All of these characteristics indicate that these cells have acquired more malignant properties than each single mutant. We discussed it in Discussion section.

On the assays from figure 2, why did you not add the crystal violet, as recommended in a typical clonogenic assay? Did you add some other dye? Please provide the reference for the protocol you used. Without a staining you might be missing some colonies;

No, we did not do any special staining which usually used for automatic calculation of colonies. We used to calculate the colonies in living cultures under microscope and in such conditions all colonies were clearly visible. We add REF of protocol for colony formation assay in soft agar which we used into Materials and Methods section..

Line 200 – I would not say controlling but rather modulating, displaying opposite effects. If PTEN is KO increased migration is observed and the opposite with ZFHX3 KO, which impairs migration; We exchanged “controlling” to “modulating”.

Is a bit contradictory how ZFHX3 KO acquire more EMT characteristics and exhibits less migratory capacity, compared to PTEN is KO. How do you explain this?

It is not contradictory. EMT is not a straightforward process of transition from epithelial to mesenchymal phenotype. In the process of EMT, cells can acquire various new properties, some of which may give them advantages in their ability to migrate, while others may give them some other advantages or none at all. For example, it is now clear that for successful migration, partial EMT is even better than full EMT; we discussed this in our article (Jolli et al., 2019, doi: 10.1016/j.pharmthera.2018.09.007). Our cells exhibit different EMT characteristics, and it is not clear which ones significantly improve migration. We believe that performing such an analysis on different tumor cells is very important to understand which features of EMT should be considered for the prognosis of the disease and the search for its treatment.

The understanding of these phenotypes in metastasis formation, in in vivo models should be suggested as discussed as a future perspective in the discussion;

We add results of analysis of acini formation in matrigel for our cells. It was shown by the work by Polizzotti et al (2012) that it is possible to estimate invasive grade of tumor according the acini morphology.

Line 331 – what do you mean by no “special type”? You mean histological subtype? Because molecular subtype, you describe between parenthesis, so is not clear to me what you mean; It is last classification of breast invasive carcinomas suggested by World Health Organization. We added the reference. The tumors named “Invasive breast carcinomas of no-special type” (IBC-NST) include a wide range of breast cancers, so we have added the molecular characteristics of the tumors studied. Now it is written: “Ten patients with invasive breast carcinoma of no special type were diagnosed and treated in the Cancer Research Institute of Tomsk NRMC (Tomsk, Russia). The histological diagnosis of breast cancer was made according to the WHO criteria (Tan et al., 2020, https://doi.org/10.1111/his.14091). All tumors were of luminal subtype, T1-3N0-2M0 stage, age range of patients was 41–67.”

Line 377 – please mention the Fiji function used for the colony counting and size analyses; Done

In 4.6. I did not quite understand how did you count the E-cadherin AJs with the cell counter plugin, could you please explain better?

We described more accurate this method in Materials and Methods.

Minor issues

We corrected all these points in text and figures

Line 71 and 72 – “in vitro and in vivo “ should be in italic. Correct throughout the text; Done

Line 91 – Please simplify this phrase e.g.: “We performed whole exome sequencing of 10 invasive mammary carcinomas.”;Done

Line 96 – replace “wondered” with something more formal, such as, pondered; Done

Legend figure 1 – Add also EGF abbreviation, like the HS; Done

Line 126 – Change “control” to parental; Done

Figure 2 – Add a more descriptive text for example in B; Done

Line 148 – “consisting of”; Done

Legend fig 3, again describe a bit more and capitalize the sentences;

Line 166 – gene-edited instead of “genome-edited”; Done

Figure 4 – Capitalize Cell Speed (in A and C), velocity (in E) and correct parenral to parental (in B and C); Done

In figure 5B is there any specific reason for the significance not being represented as in all other graphs? Uniformize with *.

5D capitalize Vimentin; Done

Line 239 – “(…) lead to an increase (…)”; Done

Line 271 – “PTEN inactivation activates” please replace one of the words; Done

Line 288 – “(…) is a dual (…)”; Done

Line 300 – gene-edited no “genome-edited”;Done

In the methods, add the cities and countries of all the companies for reagent purchase; Done

Line 348 - penicillin/streptomycin purchase company info missing; Done

 4.5. mention the concentrations of the antibodies; Done

In 4.6. the same antibodies are used? Please mention it, as well as the concentrations used; Done

In 4.7. the same antibodies are used? Please mention it, as well as the concentrations used; Done

Line 363-366 – uniformize the font; We did not find this mistake

Line 429 – add 4.8.1; Done

Line 450 – add 4.8.2.Done

Reviewer 2 Report

The study seeks to define the roles that two tumor suppressor genes play in the invasiveness of mammary carcinoma, namely PTEN and ZFHX3. As a model cell line, they use the non-tumorigenic epithelial cell line MCF10A, and performed knock-out studies, of each gene individually, as well as in combination. 

The authors performed WES on a very small cohort (n=10) of mammary carcinomas, where they found concomitant alterations of both PTEN and ZFHX3 in 20% (2 out of 10) of these tumors. 

Result section 2.1:

The authors state that "Indeed PTEN and ZFHX3 are mutated in 5.9 % and 3.0 % of tumors, respectively (cBioportal on TCGA cohort) rendering their association at random unlikely". A large number of genes can fall within this percentage range as mutated in cancers, but are seldom associated. The authors need to perform bioinformatic analyses and generate validity scores to provide quantitative substance to their claim that these two genes are associated.

Result section 2.2:

The authors state that "form islets in sparse culture and isolated cells are practically absent". The authors need to explain what is meant by "islets in sparse culture". Figure 1 does not provide a clear indication on the changes that the authors may be trying to describe. An absence of isolated single cells is more likely to be as a result of slowed proliferation than anything else - this is a likely consequence of lowering serum and growth factors. The authors should measure changes in proliferation rates of their four cell models, both in normal medium, as well as when serum is reduced. 

The authors should provide western blot (WB) results showing expression of PTEN and ZFHX3 in parental cells, and the success of KO of these two proteins in the KO cell lines. 

The authors state that "Since PTEN and ZFHX3 are two described cancer driver genes". This is entirely confusing, as both of these genes are in fact tumor suppressors, as the authors themselves have stated earlier in the paper. 

Can the authors kindly indicate how many biological replicates were performed for each of their assays (this can be indicated in the material and methods section).

Wound healing assay method: Can the authors please clarify how they made sure that the cells were migrating only, and not proliferating? Usually, a reagent, such as mytocycin-C, is included in these types of wound healing assays, especially when complete medium is used, which is the case for the experimental performed by the authors. 

Overall, this study is of average significance, and while the authors are trying hard to show some level of association between PTEN and ZFHX3, in the process of cell migration, the evidence that they provide that these two genes may be associated, is poor. It would have been of better value to analyze the role of each gene independently. Additionally, while the role that PTEN plays in inhibiting the migratory ability of the MCF10A cells can be convincingly argued through the experimental work of in this paper, the same cannot be said for ZFHX3. A crucial aspect of the work which is lacking is that the authors have not investigated the impact on proliferation. 

Author Response

Answer to reviewers.

We are very grateful to the reviewers for their careful reading of our manuscript and their valuable comments. We believe that the quality of our paper has greatly improved after taking into account the reviewers' comments. Our responses and comments can be found below (marked as red).

Reviewer 2.

Result section 2.1:

The authors state that "Indeed PTEN and ZFHX3 are mutated in 5.9 % and 3.0 % of tumors, respectively (cBioportal on TCGA cohort) rendering their association at random unlikely". A large number of genes can fall within this percentage range as mutated in cancers, but are seldom associated. The authors need to perform bioinformatic analyses and generate validity scores to provide quantitative substance to their claim that these two genes are associated.

We calculated the association of overall survival with both PTEN and ZFHX3 mutation status. The corresponding text and Figure S1 was added. Overall survival of breast cancer patients with double PTEN and ZFHX3 mutations was not calculated due to technical limitations of the cBioPortal tool.

Result section 2.2:

The authors state that "form islets in sparse culture and isolated cells are practically absent". The authors need to explain what is meant by "islets in sparse culture".

Figure 1 does not provide a clear indication on the changes that the authors may be trying to describe. An absence of isolated single cells is more likely to be as a result of slowed proliferation than anything else - this is a likely consequence of lowering serum and growth factors. The authors should measure changes in proliferation rates of their four cell models, both in normal medium, as well as when serum is reduced. 

The absence of isolated cells could be the result of both slowed proliferation and reduced ability for scattering. The ZFHX3 KO culture exhibits much more isolated cells. Also in complete medium (with addition of EGF and serum) in all of our cultures we observed much more isolated cells than in depleted conditions which indicate the increase of cell scattering as a result of EMT under EGF treatment. In main experiments in our work we used to study cell motility in depleted medium, to avoid stimulation of EMT by addition of EGF and redundant serum,  because our task was to analysed EMT caused by lack of PTEN and ZFHX3. Thus we remove from Figure 1 the column with morphology of cells in presence of EGF and horse serum, and add quantification of gaps between cells in islets (according the request of first reviewer (Fig.1C)) and data about proliferation rate for our four cell models (Fig. 1D). Also we added the picture with morphology of cells in full medium as Supplementary Figure S3

The authors should provide western blot (WB) results showing expression of PTEN and ZFHX3 in parental cells, and the success of KO of these two proteins in the KO cell lines. 

We add these results as supplementary Figure S2B. The problem is that ZFHX3 has huge size (404KDa) and thus the results concerning ZFHX3 did not look very nice, but one could see the absent of ZFHX3 in knockouted lines. 

The authors state that "Since PTEN and ZFHX3 are two described cancer driver genes". This is entirely confusing, as both of these genes are in fact tumor suppressors, as the authors themselves have stated earlier in the paper. We corrected this phrase “Since PTEN and ZFHX3 are described ,as TSGs for many of tumors we wanted to evaluate the level of transformation that these genetic alterations provide to untransformed MCF10A cells.”

Can the authors kindly indicate how many biological replicates were performed for each of their assays (this can be indicated in the material and methods section).

Each experiment was replicated independently at least three times. The exact number of examples used for statistical quantifications are noticed in legends under each figures.

Wound healing assay method: Can the authors please clarify how they made sure that the cells were migrating only, and not proliferating? Usually, a reagent, such as mytocycin-C, is included in these types of wound healing assays, especially when complete medium is used, which is the case for the experimental performed by the authors.  We did not use complete medium in these experiments

We did these experiments in condition of depleted medium and absence of EGF, each experiment was done for 10 hours, so in these conditions impact of proliferation could not be significant. We also add the results of analysis of proliferation intensity of our cells in depleted medium and in presence of horse serum and EGF for 16 hours (Figure 1D), and did not reveal any significant differences in rate of proliferation between our cells. Thus the proliferation impact could not be significant in wound healing in our condition.  We added this explanation to text

Overall, this study is of average significance, and while the authors are trying hard to show some level of association between PTEN and ZFHX3, in the process of cell migration, the evidence that they provide that these two genes may be associated, is poor.

We have discovered these two genes as drivers in one tumor between 10 investigated and together with recent data showed that combined inactivation of ZFHX3 and PTEN was found to drive the progression of prostate cancer in a mouse model (Sun et al, 2015) it became the initial point of our study

It would have been of better value to analyze the role of each gene independently. Additionally, while the role that PTEN plays in inhibiting the migratory ability of the MCF10A cells can be convincingly argued through the experimental work of in this paper, the same cannot be said for ZFHX3. A crucial aspect of the work which is lacking is that the authors have not investigated the impact on proliferation. 

We added an analysis of proliferation in our cells, see above. We analyzed how individual PTEN and ZFHX3 knockouts and their combination contribute to cell migration and EMT trait acquisition, and showed that we cannot explain the increase in cell migration capacity as a result of simply summing EMT traits. We think that such analysis is very important because it could give ideas as to which combination of EMT features lead to acquisition of metastatic phenotype.

Reviewer 3 Report

In this study, Dayoub et al. using MCF10A cell line investigated the effects of a double KO of the PTEN and ZFHX3 gene on cell migration and its potential impact on EMT in breast cancer.

The authors show that both single and double KO of PTEN and ZFHX3 genes in MCF10A affects migration and the anchorage-independent growth capacity of the cells, cultured in EGF-free and serum-reduced media. Analyzing the morphology of E-cadherin-based AJs, they describe significant differences between modified and parental cells. Next, they show that inactivation of the ZFHX3 gene is closely associated with an increased number of vimentin-positive cells in culture, and this number further increases in the case of double KO. To sum up, the authors state that the cells with double KO show specific motility. different from that observed in single knockouts, characterized by reduced coordination in collective migration, high levels of vimentin and mature linear AJs.

The article is well written and easy to read. Methods used in the work are comprehensively described. The obtained results are correctly presented. However, I have a few comments.

Major:

First, there is a lack of a blot to provide evidence that the authors are working with cells lacking proteins encoded by targeted genes. Also, nothing is known about the expression levels of PTEN and ZFHX3 in paternal MCF10A.

Based on the observation of changes in vimentin levels, the authors conclude that double KO induces the EMT process in cells. In the absence of changes in E-cadherin levels, analysis of only one marker is insufficient. The authors should determine the levels of other EMT markers, such as N-cadherin or selected keratins. It must be remembered that MCF10A cells are known to show differential expression of EMT marker genes depending on whether the cells are cultured under sparse or confluent conditions. The authors do not mention this.

Similarly, they do not discuss the effect of proliferation on wound healing assay. It is known that ZFHX3 regulates the expression of genes related to the cell cycle. Consequently, MTT assays should be performed for all cell lines.

The authors believe that inactivation of the ZFHX3 gene in MCF10A PTEN KO cells induces a more advanced stage of EMT, although at the same time it slows down the rate of cell migration. This is not typical of EMT. To confirm that the observed changes are associated with greater invasive potential, it is necessary to compare the ability of individual MCF10A lines to form mammospheres and invade in 3D Matrigel cell culture.

Minor:

Fig.S1 and S2 are not included in the Supplementary materials.

In my opinion, this manuscript in its present form is not suitable for publication.

Author Response

Answer to reviewers.

We are very grateful to the reviewers for their careful reading of our manuscript and their valuable comments. We believe that the quality of our paper has greatly improved after taking into account the reviewers' comments. Our responses and comments can be found below (marked as red).

Reviewer 3.

First, there is a lack of a blot to provide evidence that the authors are working with cells lacking proteins encoded by targeted genes. Also, nothing is known about the expression levels of PTEN and ZFHX3 in paternal MCF10A.

We add this blot as Supplementary Figure S2B. The problem was that ZFHX3 has huge size (404KDa) and thus the results concerning ZFHX3 did not look very nice, but one could see the absent of ZFHX3 in knockouted lines.

Based on the observation of changes in vimentin levels, the authors conclude that double KO induces the EMT process in cells. In the absence of changes in E-cadherin levels, analysis of only one marker is insufficient. The authors should determine the levels of other EMT markers, such as N-cadherin or selected keratins.

We provide the analysis of additional EMT markers such as N-cadherin, cytokeratins 8 and 5,6, as well as p-cadherin, see Fig 5.

It must be remembered that MCF10A cells are known to show differential expression of EMT marker genes depending on whether the cells are cultured under sparse or confluent conditions. The authors do not mention this.

 We added this to section Discussion

Similarly, they do not discuss the effect of proliferation on wound healing assay. It is known that ZFHX3 regulates the expression of genes related to the cell cycle. Consequently, MTT assays should be performed for all cell lines.

We did wound healing assay in condition of depleted medium and absence of EGF, each experiment was done for 10 hours, so in these conditions impact of proliferation could not be significant. We also add the results of analysis of proliferation intensity for our cells in depleted medium and in presence of horse serum and EGF for 16 hours (Figure 1D). To estimate cell proliferation activity we quantified  the percentage of cells in S-phase which was scored as the ratio of EdU-positive nuclei / DAPI-stained nuclei. These quantifications did not reveal any significant differences in rate of proliferation between our cells. Thus the proliferation impact could not be significant in wound healing in our condition. 

The authors believe that inactivation of the ZFHX3 gene in MCF10A PTEN KO cells induces a more advanced stage of EMT, although at the same time it slows down the rate of cell migration. This is not typical of EMT. To confirm that the observed changes are associated with greater invasive potential, it is necessary to compare the ability of individual MCF10A lines to form mammospheres and invade in 3D Matrigel cell culture.

EMT is not a straightforward process of transition from epithelial to mesenchymal phenotype. In the process of EMT, cells can acquire various new properties, some of which may give them advantages in their ability to migrate, while others may give them some other advantages or none at all. For example, it is now clear that for successful migration, partial EMT is even better than full EMT; we discussed this in our article. Our cells exhibit different EMT characteristics, and it is not clear which ones significantly improve migration. We believe that performing such an analysis on different tumor cells is very important to understand which features of EMT should be considered for the prognosis of the disease and the search for its treatment. We added the experiments with mammospheres in 3D Matrigel (section 2.5 and Supplementary Figure S4).

Minor:

Fig.S1 and S2 are not included in the Supplementary materials.

Sorry, during first submission these Figures were in folder Figures. We combined all Supplementary materials in separate folder.

Round 2

Reviewer 1 Report

My questions have been addressed. Some English typos are present and need one last proof reading before publication, e.g. N-Cadherin in line 261.

Author Response

Thank you very much for reading our manuscript carefully. We have corrected the typos that you noticed, as well as some others in the text, and we hope that our corrected manuscript now looks better and is ready for publication. Please find our corrected manuscript in the attachment.

Reviewer 3 Report

In the revised manuscript, the authors have addressed all my concerns and made the necessary changes to the manuscript in a satisfactory way.

Author Response

Dear Reviewer,

Thank you very much for reading our manuscript carefully. We have corrected some typos in the text, and we hope that our corrected manuscript now looks better and is ready for publication. Please find our corrected manuscript in the attachment.
